# Glucose-to-Lymphocyte Ratio (GLR) as an Independent Prognostic Factor in Patients with Resected Pancreatic Ductal Adenocarcinoma—Cohort Study

**DOI:** 10.3390/cancers16101844

**Published:** 2024-05-11

**Authors:** Su-Hyeong Park, In-Cheon Kang, Seung-Soo Hong, Ha-Yan Kim, Ho-Kyoung Hwang, Chang-Moo Kang

**Affiliations:** 1Division of Hepatobiliary and Pancreatic Surgery, Department of Surgery, International St. Mary’s Hospital, Catholic Kwandong University College of Medicine, Incheon 22711, Republic of Korea; kzpgg@ish.ac.kr; 2Department of Surgery, CHA Bundang Medical Center, CHA University, Seongnam 13497, Republic of Korea; vorbote@cha.ac.kr; 3Division of Hepatobiliary and Pancreatic Surgery, Department of Surgery, Yonsei University College of Medicine, Seoul 03722, Republic of Korea; hongss@yuhs.ac (S.-S.H.); drhhk@yuhs.ac (H.-K.H.); 4Pancreatobiliary Cancer Clinic, Severance Hospital, Seoul 03722, Republic of Korea; 5Department of Biomedical System Informatics, Yonsei University College of Medicine, Seoul 03722, Republic of Korea; hykim1213@yuhs.ac

**Keywords:** pancreatic cancer, pancreatic neoplasm, pancreatic ductal adenocarcinoma, marker, survival

## Abstract

**Simple Summary:**

This study suggests that GLR, along with other factors like CA 19-9 levels and symptoms, can help predict long-term survival outcomes in PDAC patients following surgical resection, aiding in the identification of high-risk individuals who may benefit from closer monitoring or more aggressive treatment approaches.

**Abstract:**

**Background:** We retrospectively evaluated the usefulness of an elevated glucose-to-lymphocyte ratio (GLR) as a sensitive prognostic biomarker of disease-specific survival in 338 patients who underwent surgical resection of pancreatic ductal adenocarcinoma (PDAC). **Methods:** The optimal GLR cutoff value was determined using the method of Contal and O’Quigley. Patient demographics, clinical information, and imaging data were analyzed to identify preoperative predictors of long-term survival outcomes. **Results:** Elevated GLR correlated significantly with aggressive tumor biologic behaviors, such as a high carbohydrate antigen (CA) 19-9 level (*p* = 0.003) and large tumor size (*p* = 0.011). Multivariate analysis identified (1) GLR > 92.72 [hazard ratio (HR) = 2.475, *p* < 0.001], (2) CA 19-9 level > 145.35 (HR = 1.577, *p* = 0.068), and (3) symptoms (*p* = 0.064) as independent predictors of long-term, cancer-specific survival. These three risk factors were used to group patients into groups 1 (0 factors), 2 (1–2 factors), and 3 (3 factors), which corresponded to significantly different 5-year overall survival rates (50.2%, 34.6%, and 11.7%, respectively; *p* < 0.001). **Conclusions:** An elevated preoperative GLR is associated with aggressive tumor characteristics and is an independent predictor of poor postoperative prognosis in patients with PDAC. Further prospective studies are required to verify these findings.

## 1. Introduction

Pancreatic ductal adenocarcinoma (PDAC) is one of the most aggressive types of cancer, with a 5-year survival rate of <10% [1]. For patients with resectable PDAC, previous studies have identified several prognostic factors as significant determinants of reduced OS in patients with resectable PDAC [2,3,4,5,6]. However, most of these factors can only be determined via postoperative pathology, underscoring the need for a new preoperative system that can predict the outcomes of patients with PDAC. 

In recent studies, systematic inflammatory markers have been demonstrated to be useful preoperative predictors of the progression and prognosis of various cancers, including PDAC [7,8,9,10]. Bhatti et al. [11] demonstrated the significance of the neutrophil-lymphocyte ratio (NLR) as a prognostic indicator in patients with resected PDAC. The increasing recognition of these biomarkers and their prognostic relevance with respect to survival has led to the recent proposal of predictive nomograms based on combinations of markers [12,13].

Interestingly, lymphocyte count is one of the most frequently included parameters in combination biomarkers. A decreased lymphocyte count indicates host immunosuppression, which may adversely affect the antitumor response of the host [14,15,16]. Fogar et al. [16] concluded that a decreased lymphocyte count was the main immunologic change associated with advanced pancreatic cancer, suggesting the importance of immune surveillance and editing in PDAC. A recent study [17] suggested that host immunity represented by the preoperative total lymphocyte count at the diagnostic stage influences the oncologic outcome in resected left-sided pancreatic cancer. 

Numerous other investigations have determined an association between diabetes mellitus (DM) and survival prognosis in PDAC patients [18,19,20,21,22]. In addition, DM is a well-known risk factor for pancreatic cancer [23,24] and is more frequently associated with pancreatic cancer than with other malignancies [25,26]. In recent studies, new-onset DM (i.e., a diagnosis of DM within 24 months before the diagnosis of PDAC) was identified as a particular marker of an aggressive tumor biology and an early sign of PDAC [25,27,28]. Accordingly, Chari et al. [25,29] suggested that a specific biomarker for pancreatic cancer-induced diabetes may enable pancreatic cancer screening in cases of new-onset DM, while Bartosch et al. [30] identified newly diagnosed DM as an initial screening tool for asymptomatic early pancreatic cancer. Takeda et al. [31] reported that asymptomatic PDAC, which is generally detected at an earlier stage and thus has a higher chance of respectability, is associated with better long-term outcomes than symptomatic PDAC. This finding suggests the potential benefits of screening programs for the early detection of PDAC in selected high-risk populations and underscores the need for careful estimation of prognosis during the initial presentation of a patient with PDAC. Luo et al. [32] and Yang et al. [33] each reported fasting blood glucose levels as prognostic factors in non-small cell lung cancer and colorectal cancer, respectively, supporting the role of preoperative hyperglycemia as a prognostic factor in various cancers. The role of preoperative serum glucose-to-lymphocyte ratio as a prognostic factor in pancreatic cancer is reported in the study by Zhang et al., which also presents a nomogram based on this ratio [34].

Based on these previous studies, we designed a preoperative prognostic biomarker called the glucose-lymphocyte ratio (GLR) and assessed the prognostic effect of GLR in pancreatic cancer patients undergoing upfront surgery. Furthermore, we developed a new scoring system by incorporating additional predictive factors. We assumed that a decreased lymphocyte count would suggest a poor host immune status, while an elevated preoperative glucose level (i.e., hyperglycemia) would suggest aggressive tumor biology. Consequently, we hypothesized that an elevated GLR might be a sensitive prognostic biomarker in patients with PDAC. In a recent paper published by Zhong et al. [35] in 2020, the concept of GLR in unresectable PDAC and its significance as a prognostic factor also supported this assumption. Therefore, in this study, we aimed to verify the potential association of GLR with long-term oncologic outcomes, especially disease-specific survival (DSS), in patients with PDAC after curative-intent surgical resection. 

We present the following article in accordance with the TRIPOD guidelines checklist.

## 2. Methods

### 2.1. Patient Selection

The medical records of patients with pancreatic cancer who underwent curative surgical resection at Severance Hospital, Seoul, Korea, between March 2002 and December 2020 were reviewed. Only patients with pathologically diagnosed PDAC were included in this study. To minimize the influence of potential confounders on preoperative immunologic function, the following exclusion criteria were applied: (1) a personal history of prior malignancy, autoimmune disease, or transplantation; (2) synchronous or metachronous malignancy; and (3) treatment with preoperative neoadjuvant chemotherapy. The final cohort included 338 patients.

### 2.2. Data Collection

Demographic, clinical, and pathological data were collected from the electronic medical records of all patients. Personal and family medical histories, and clinical presentations at diagnosis, were carefully reviewed. Weight loss was defined as an unintentional loss of >5 kg reported at the initial diagnosis, and jaundice was defined as a total bilirubin level of >3 mg/dL. The presenting symptoms include abdominal pain, indigestion, anorexia, and vomiting. For all patients, preoperative diagnostic evaluations and tumor staging were based on the results of computed tomography and/or magnetic resonance imaging. Information regarding tumor characteristics (size and location) was collected from preoperative imaging studies. Preoperative laboratory tests (complete blood counts with differentials, FBG, and Carbohydrate Antigen 19-9 (CA 19-9) levels) were conducted according to the guidelines for anesthesia after 8 h of fasting. In our institution, all patients subjected to general anesthesia were evaluated for operability by conducting laboratory examinations, including a complete blood count (CBC) and serum analysis (SMA), valid only within 1 month before surgery. For patients with long-standing diabetes, glucose levels measured during medication or insulin therapy within one month before surgery were used as test results. Lymphocytes and glucose levels were measured on the same day; however, the date may be different if it was necessary that laboratory findings were retested (e.g., low hemoglobin). Laboratory findings for which the result is not likely to change even after repeated tests were not retested. For repeated measurements, the latest laboratory test performed preoperatively was used as the reference. Patients with symptoms of jaundice before surgery underwent endoscopic procedures for stent insertion or drainage, followed by laboratory tests including tumor markers (CA 19-9, CEA). Intraoperative findings (surgery type) and pathological data were also extracted from medical records. The pathological tumor stages were categorized according to the 8th American Joint Committee on Cancer Staging (AJCC) Manual [36]. The primary endpoint was disease-specific survival (DSS), defined as the time interval from the date of surgery until pancreatic cancer-related death. DSS only included deaths due to pancreatic cancer, excluding deaths due to other diseases or complications, for a more accurate evaluation of the relationship between glucose-lymphocyte ratio (GLR) and prognosis of pancreatic cancer by excluding other factors affecting survival. 

### 2.3. Ethics Approval and Consent to Participate

This study was approved by the Institutional Review Board (IRB) of Yonsei University College of Medicine (IRB number: 2018-3279-001). Based on South Korean regulations, retrospective studies without any additional therapy or monitoring procedures do not require formal written consent from the patients. This study was conducted in accordance with the principles of the Declaration of Helsinki.

### 2.4. Definition of Systemic Inflammatory Markers and Determination of Best Cut-Off Points

In this study, several combinations of systemic inflammatory markers were included in the analysis of the potential prognostic factors. GLR was defined as the FBG level divided by the lymphocyte count. The NLR was defined as the neutrophil count divided by the lymphocyte count. The PLR (Platelet–Lymphocyte ratio) was defined as the platelet count divided by the lymphocyte count. The best cut-off points for these markers were determined using the statistical methods derived by Contal and O’Quigley, which calculated the maximizing HR based on log-rank tests and estimated the best cut-off value [37]. The same method was applied to the CA 19-9 levels and tumor size imaging. GLR was dichotomized at 92.72, *p*-value < 0.001, CA19-9 at 145.35, *p*-value 0.028, and image size at 2.0 cm of *p*-value 0.060.

### 2.5. Statistical Analysis 

Categorical data were summarized and reported as frequencies and percentages and were compared using the chi-square or Fisher’s exact test, as indicated. Continuous variables were summarized and reported as means and standard deviations or as medians and interquartile ranges and were compared using Student’s *t*-test or the Mann–Whitney U test (if the variables had an abnormal distribution). Survival curves were calculated using the Kaplan–Meier method and compared using the log-rank test. Factors influencing DSS were identified using univariate and multivariate Cox proportional hazards regression models. All variables found to be significant (*p* < 0.05) in the univariate analysis were entered into a step-down Cox proportional hazards regression analysis. HRs and 95% CIs were obtained for all the regressions. All statistical analyses were performed using SPSS version 27 (IBM Corp., Armonk, NY, USA) and SAS version 9.2 (SAS Institute Inc., Cary, NC, USA). Statistical significance was set at *p* < 0.05.

## 3. Results

### 3.1. Baseline Characteristics

This study included 338 patients with histopathologically confirmed PDAC, of whom 201 (59.5%) and 137 (40.5%) were male and female, respectively. The median patient age was 64 years (mean age, 63.6 ± 9.65 years), and the median tumor size was 2.5 cm (mean size, 2.87 ± 1.20 cm). A total of 189 (55.9%) patients underwent pancreaticoduodenectomy (PD) or pylorus-preserving (PP) PD, 138 (40.8%) underwent distal pancreatectomy (DP) and 11 (3.3%) underwent total pancreatectomy (TP). Regarding margin status, 292 (86.4%) and 46 (13.6%) patients underwent R0 or R1 resection, respectively. The median follow-up duration was 24.07 months. Adjuvant chemotherapy was administered to 272 patients (80.5%) and initiated within six weeks after surgery. The median GLR value was 82.47 (range, 9.33–480.95), and the median NLR and platelet–lymphocyte ratio (PLR) values were 2.07 (range, 0.41–22.92) and 137.20 (range, 1.26–594.44), respectively. The general characteristics of the 338 patients are presented in Table 1.

### 3.2. Preoperative Clinical Parameters Used to Estimate Long-Term Oncologic Outcomes

The median overall survival duration was 29.4 months (95% CI: 24.1–34.8), and the 5-year overall survival rate was 14.5%. In the present study, the statistical analysis included only clinically available preoperative parameters, which were broadly divided into three categories: (1) demographic data (age, sex, smoking history, DM status, clinical presentation at diagnosis); (2) laboratory data [neutrophil, platelet, and lymphocyte counts; fasting blood glucose (FBG), carbohydrate antigen (CA) 19-9 levels; GLR, NLR, and PLR]; and (3) imaging data (tumor size and location).

The statistically determined best cutoff points were as follows; 92.72 GLR, 1.06; NLR, 133.96; PLR, 145.35 U/mL; CA 19-9, and tumor size, 2 cm. In univariate analysis, new-onset diabetes mellitus, the presence of symptoms, glucose level, GLR > 92.72, CA19-9 > 145.35 U/mL, and imaged tumor size >2 cm were identified as statistically significant predictors of DSS (*p* < 0.05). However, other combination biomarkers such as NLR and PLR were not identified as significant predictors of DSS prognosis in patients with resected PDAC.

In a multivariate regression analysis, a GLR > 92.72 [hazard ratio (HR) = 2.475, 95% confidence interval (CI): 1.502–4.078, *p* < 0.001] was identified as an independent predictors of long-term DSS prognosis, while CA19-9 > 145.35 U/mL (HR = 1.577, 95% CI: 0.966–2.572, *p* = 0.068) and the presence of symptoms (HR = 1.582, 95% CI: 0.974–2.568, *p* = 0.064) were expected to be marginally significant independent predictors of DSS prognosis (Table 2, Figure 1). Serum glucose was designated as an overlapping factor with GLR and was excluded from the multivariate analysis.

### 3.3. Clinicopathologic Characteristics According to Preoperative GLR

The patients were divided into low (≤92.72) and high (>92.72) groups. A comparison of the clinicopathological characteristics of the patients with low and high GLR values is shown in Table 3. Of the 338 patients, 203 (60.1%) had low GLR and 135 (39.9%) had high GLR. Compared with patients with a low GLR, those with a high GLR had a significantly higher DM prevalence (*p* < 0.001), higher CA 19-9 level (*p* = 0.003), and larger tumor size (*p* = 0.011). Age, sex, pathological results, postoperative complications, adjuvant chemotherapy, and type of chemotherapy regimen did not differ significantly between the two groups. In our hospital, adjuvant FOLFIRINOX chemotherapy has been implemented since 2017. In terms of the type of operation, PPPD was the most common in both groups and was significantly more frequently performed in the high GLR group.

### 3.4. Survival Stratified by the Number of Prognostic Factors

After a median follow-up of 24.07 months (range, 1.00–138.2 months), 147 patients (43.5%) died from pancreatic cancer-related disease. In the entire cohort, the 1-, 3-, and 5-year DSS rates were 83.1%, 43.9%, and 33.1%, respectively. We devised a preoperative survival prediction system using three independent prognostic factors: GLR, CA 19-9 level, and presence of symptoms. All patients were categorized into three risk groups according to the number of independent prognostic factors (Table 4). Group 1 (low-risk group) did not have any prognostic factors (*n* = 56, 16.5%). Group 2 (intermediate-risk group) was defined as those with one or two prognostic factors (*n* = 224, 66.3%). Group 3 (high-risk group) was defined as those with all three prognostic factors (*n* = 58, 17.2%). The survival estimation curves of the patients stratified by prognostic factors are shown in Figure 2. The DSS rate worsened as the number of independent prognostic factors increased, with 5-year rates of 50.2%, 34.6%, and 11.2% in groups 1, 2, and 3, respectively (Group 1 vs. Group 3, *p* < 0.001; Group 1 vs. Group 2, *p* = 0.006; Group 2 vs. Group 3, *p* < 0.001).

### 3.5. Relation between Risk Group and DSS in Adjuvant Chemotherapy

The disease-specific survival (DSS) in cases where adjuvant chemotherapy was not administered had a median value of 19.30 months (95% CI 8.91–29.69 months), whereas in cases where adjuvant chemotherapy was administered, the DSS had a median value of 32.00 months (95% CI 24.88–39.12 months). For Group 3 (high-risk group) that did not undergo adjuvant chemotherapy, the DSS was a median of 13.57 months (95% CI 5.14–22.00 months), and for Group 3 in the group that received adjuvant chemotherapy, the DSS was a median of 17.67 months (95% CI 12.00–23.34 months) (*p* < 0.001) (Figure 3).

## 4. Discussion

According to recently published studies related to GLR, it has been confirmed that GLR can be used as a prognostic factor in various diseases [38,39,40]. Cai et al. [38] concluded that there is a close association between mortality and GLR in patients with sepsis, and Chen et al. [39] demonstrated the utility of GLR in predicting outcomes in peritoneal dialysis patients. Additionally, it has been confirmed that GLR can be used to predict the prognosis of patients with acute pancreatitis [40]. In this study, we investigated the usefulness of the preoperative glucose-to-lymphocyte ratio (GLR) as a predictor of long-term oncologic outcomes in resected PDAC and determined that the combination of a high GLR, high CA 19-9 level, and the presence of symptoms could be used to preoperatively predict high-risk patients. Prognostic factors are crucial for the administration of personalized medicine. However, most prognostic factors for PDAC are defined based on pathological examination after surgical extirpation, and preoperatively available prognostic parameters remain particularly controversial. In this context, we previously reported a preoperative radiological image-based morphologic analysis to predict the long-term survival of resected PDAC [41].

As cancer progression relies on complex interactions between the tumor and the host immune system [8,9], a state of immune suppression may reduce the host antitumor response and thus have deleterious effects on survival [42]. Accordingly, Zheng et al. [43] explained the pathogenesis of PDAC in terms of immunology based on substantial evidence supporting the association of host factors (e.g., weight loss, poor performance status, and systemic inflammatory response) with disease activity in patients with advanced cancer. van Wijk et al. [10] noted C-reactive protein (CRP) as a marker of cancer activity and albumin as a marker of nutritional and immune status of the host in resected pancreatic cancer patients. They mentioned that elevated serum CRP levels might be caused by tumor necrosis or local tissue damage and upregulated in response to elevated interleukin-6 levels, which promotes tumor growth by inducing multiple signaling pathways. Additionally, high interleukin-6 levels produced by cancer cells inhibit the synthesis of albumin, resulting in hypoalbuminemia. In this regard, a decreased lymphocyte count [16,17] and an elevated NLR [11], PLR [44], and C-reactive protein-to-albumin ratio (CAR) were identified as independent predictors of poor survival among patients with PDAC. In contrast to these findings, our analysis did not identify well-known combination markers such as NLR and PLR as independent factors associated with DSS in patients with PDAC after surgical resection.

Interestingly, in our previous study, we identified GLR as an independent prognostic factor in patients with stage T2 gallbladder cancer [45], thus suggesting the potential application of this combination factor as a prognostic biomarker in other malignancies. According to recent studies, regarding renal cell carcinoma, it has been confirmed that GLR acts as an independent factor in predicting the prognosis of patients who underwent laparoscopic nephrectomy [46]. An elevated glucose levels have been shown to be strongly associated with cancer-related survival in previous studies of other cancers [33,47,48]. As recent research by Zhang et al. [34] has demonstrated, GLR also functions as a prognostic factor in pancreatic cancer. Since the role of glucose in cancer cells was first described in 1956, aerobic glycolysis has become a hallmark of cancer biology [49]. Recent studies have suggested that high 18F-fluorodeoxyglucose uptake into tumors is associated with poor survival outcomes in patients with PDAC [50,51]. Thus, this increased glycolysis activity can be hypnotized as an alternative parameter to estimate aggressive tumor biology. In relation to this, our previous research on the prognosis prediction of patients before surgery, specifically regarding the survival and recurrence rates of new-onset diabetes mellitus (DM) and pancreatic cancer, revealed that the survival rate in the new-onset DM pancreatic cancer group was lower and the recurrence rate was higher compared to both the non-DM pancreatic cancer group and the longstanding DM pancreatic cancer group [28]. This is attributed to the association between high glucose levels and aggressive tumor biology, which is considered an early sign of pancreatic cancer. Our study also confirmed a higher prevalence of DM in the high glucose-to-lymphocyte ratio (GLR) group, providing a reason for utilizing GLR along with immune markers represented by lymphocytes as indicators for preoperative prognosis prediction. As a result, the preoperative GLR level seems to have a high potential to serve as a marker for predicting aggressive tumor behavior in pancreatic cancer. In relation to this, it was observed that, as the preoperative level of glucose-to-lymphocyte ratio (GLR) increased, there was a presence of lymph node metastasis (*p* = 0.038).

Here, we determined that the preoperative GLR could potentially be used to identify aggressive tumor biological behavior, given its significant associations with known markers of aggressive behavior, such as a high CA 19-9 level and large tumor size (Table 3). Additionally, we observed a significantly higher prevalence of DM among patients with a high GLR. Evidence suggests that hyperglycemia and impaired glycemic control are associated with immune dysfunction (e.g., lymphopenia) [52,53]. Our study showed a significant correlation between hyperglycemia and DSS (*p* = 0.007), but there was no association between new-onset DM (*p* = 0.117) or long-standing DM (*p* = 0.258) and DSS (Table 2). Regarding the prediction of cancer recurrence sites, including local recurrence and systemic recurrence, there was no significant difference observed between the low GLR group and high GLR group. No significant differences were observed in the rates of local recurrence, systemic recurrence, and both (local and systemic) recurrence between the low and high GLR groups (29.5%, 50.0%, 20.5%, and 24.7%, 56.8%, and 18.5%, respectively, *p* = 0.639) (Table 3). These results suggest that various factors beyond GLR may influence recurrence. Additional research is needed, as various factors can influence the recurrence sites.

In addition to the GLR, we further identified the preoperative CA 19-9 level and the presence of preoperative symptoms as independent prognostic factors, well-known prognostic factors in patients with resected PDAC, and demonstrated significant differences in survival using these three factors (Table 2). The combination of GLR, CA 19-9, and the presence of symptoms appears to be useful for preoperatively predicting the oncologic outcomes of patients with resected PDAC. The timing of surgery is another important factor for successful outcome. Based on our individual risk system, this treatment strategy would best enable patient-specific care and treatment optimization. In a real-world clinical setting, the question of “whether to treat” should be answered prior to “which regimen to use,” although the former requires the use of prognostic factors that would identify patients who might (or might not) benefit from treatments such as chemotherapy. Several relevant recent reports that considered whether neoadjuvant treatment (NAT) would improve oncologic outcomes have underscored the need to carefully balance surgical decisions regarding high-risk patients [54,55,56], The ability to accurately predict long-term oncologic outcomes before surgery can help select suitable candidates for NAT. In particular, when high-risk patients are predicted along with a high risk of surgical complications due to comorbidity, we can cautiously consider NAT rather than upfront surgery. Similarly, adjuvant chemotherapy can be planned and implemented at an appropriate time by predicting high-risk groups using the GLR. In our study, patients who underwent adjuvant chemotherapy showed a higher survival rate than those who did not (median 19.30 months vs. 32.00 months). In the comparison of the two groups, there was no significant difference in survival based on the administration of adjuvant chemotherapy in Group 1 (low risk). However, in Group 2 (intermediate risk group) and above, a difference in survival was observed (median 18.60 months vs. 35.73 months in Group 2, *p* = 0.005; median 13.57 months vs. 17.67 months in Group 3, *p* < 0.001). Postoperative adjuvant chemotherapy is known to improve survival in various studies, and through our research [57,58,59,60], it can be considered that adjuvant chemotherapy should be more actively applied to high-risk patients. Additional research on the analysis of regimens based on risks is required. 

GLR is known to play a significant role as a prognostic factor in various cancer types, including NSCLC, CRC, breast cancer, gastric cancer, kidney cancer, liver cancer, esophageal cancer, and melanoma [61]. In our study, unlike research on other cancer types, we expect that GLR, which has a direct relationship with the pancreas that secretes insulin to regulate serum glucose levels, will play a more effective and accurate role in prognostic prediction for pancreatic cancer patients.

Our study has some limitations. First, this was a single-institution retrospective study, for which most data were extracted from medical records; therefore, missing data and selection bias are inevitable. Second, the patients received heterogeneous treatments, including pancreaticoduodenectomy or pylorus preserving pancreaticoduodenectomy, distal pancreatectomy, and total pancreatectomy, which might have had differential influences on survival. Third, inflammatory markers may be altered by a wide range of conditions, including acute inflammation. However, since we performed elective surgery after all acute inflammations, such as cholangitis or pancreatitis, were resolved in the process of evaluating operability, this effect is likely to be minimized. Fifth, serum glucose levels can vary depending on factors such as the timing of measurement and the patient’s condition, regardless of each patient’s glucose tolerance. Correction of serum glucose data in patient groups using HbA1c measurement could be considered; however, our research institution does not routinely measure HbA1c, resulting in significant missing data, which precluded such comparisons. Therefore, the results should be validated in a large-scale prospective clinical study based on external data. Despite these limitations, preoperative GLR is a simple, non-invasive, and easily determined biomarker. 

## 5. Conclusions

In conclusion, an elevated preoperative GLR is correlated with aggressive tumor biology and constitutes a negative independent predictor of the prognosis of patients with resected PDAC. Our findings should be validated through well-designed larger prospective studies.

## Figures and Tables

**Figure 1 cancers-16-01844-f001:**
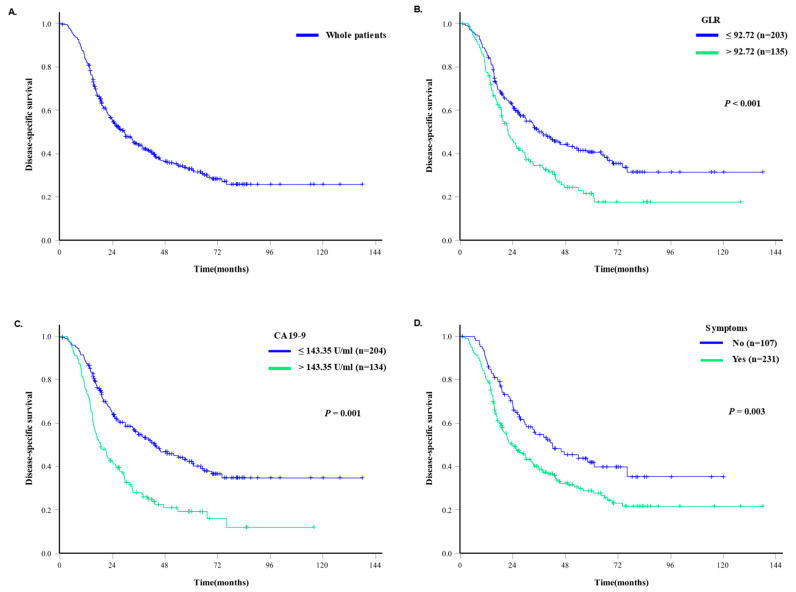
Kaplan–Meier survival curves of patients who underwent surgery for PDAC. (**A**). Disease-specific survival outcomes of all 338 patients who underwent surgical resection for PDAC. (**B**). Disease-specific survival outcomes according to the glucose-to-lymphocyte ratio (GLR). (**C**). Disease-specific survival outcomes according to carbohydrate antigen (CA) 19-9 levels. (**D**). Disease-specific survival outcomes according to the presence of symptoms.

**Figure 2 cancers-16-01844-f002:**
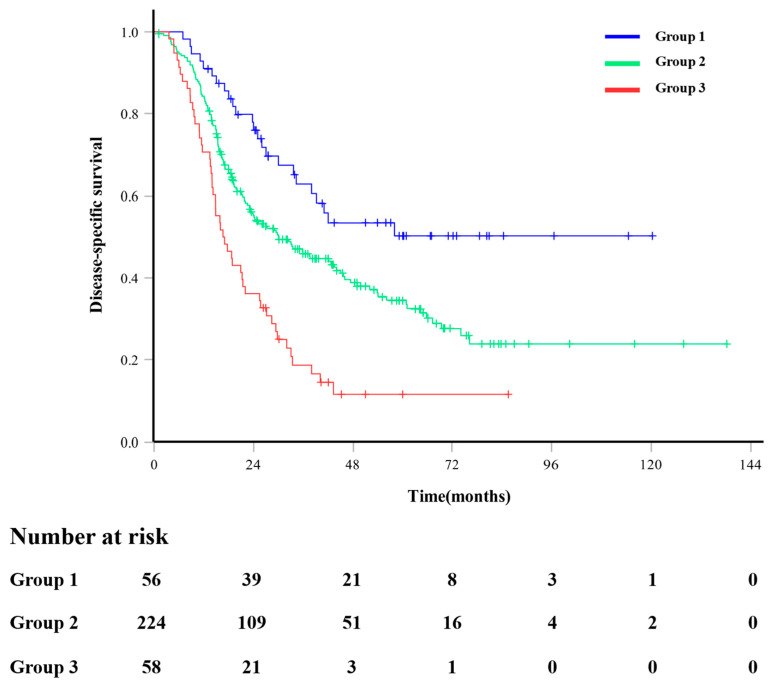
Disease-specific survival outcomes according to the number of independent prognostic factors. Group 1 included patients with no independent prognostic factors. Group 2 included patients with 1 or 2 independent prognostic factors. Group 3 included patients with three independent prognostic factors.

**Figure 3 cancers-16-01844-f003:**
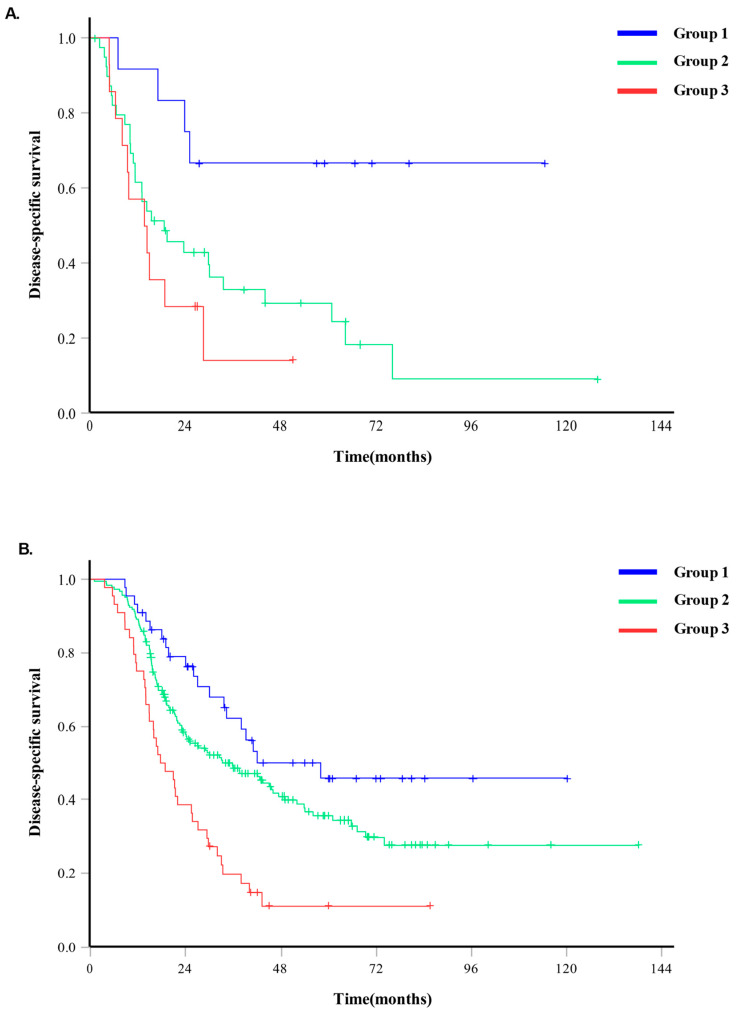
The difference in disease-specific survival according to the administration of adjuvant chemotherapy in each risk group. (**A**). When adjuvant chemotherapy was not administered, the disease-specific survival was 19.30 months (95% CI 8.91–29.69 months). (**B**). When adjuvant chemotherapy was administered, the disease-specific survival had a median value of 32.00 months (95% CI 24.88–39.12 months).

**Table 1 cancers-16-01844-t001:** Baseline characteristics of patients with resected pancreatic ductal adenocarcinoma.

Baseline Characteristic	Total (N = 338)
Age (years, range)	63.6 ± 9.65(64, 33–82); mean (median)
Sex (M/F)	201 (59.5%)/137 (40.5%)
DM (None/New-onset/Long-standing)	180 (53.3%)/76 (22.5%)/82 (24.3%)
Symptom	231 (68.3%)
Jaundice	84 (24.9%)
Weight loss	62 (18.3%)
Surgery type (PD/PPPD/DP/TP)	13 (3.8%)/176 (52.1%)/138 (40.8%)/11 (3.3%)
CA 19-9, U/mL	492.33 ± 1671.95(82, 0.1–20,000); mean (median)
Serum glucose level, mg/dL	158.13 ± 87.50 (127, 42–556); mean (median)
Hyperglycemia (FBG > 110 mg/dL)	229 (67.8%)
Neutrophilia (>7.0/mL × 10^6^/mL)	22 (6.5%)
Thrombocytosis (>400/mL × 10^6^/mL)	20 (5.9%)
Lymphocytopenia (<1.1/mL × 10^6^/mL)	47 (13.9%)
GLR	105.42 ± 76.47 (82.47, 9.33–480.95); mean (median)
NLR	2.60 ± 2.25(2.07, 0.41–22.92); mean (median)
PLR	157.18 ± 81.22 (137.20, 1.26–594.44); mean (median)
Tumor location (Head/Body + Tail)	198 (58.6%)/140 (41.4%)
Pathologic tumor size, cm (range)	2.87 ± 1.20 (2.50, 0.90–7.70); mean (median)
Tumor differentiation (Well/Moderate/Poor)	42 (12.4%)/254 (75.1%)/42 (12.4%)
R status (R0/R1)	292 (86.4%)/46 (13.6%)
TNM stage (I//IIA/IIB/III/IV, AJCC, 8th edition)	122 (36.1%)/13 (3.8%)/101 (29.9%)/97 (28.7%)/5 (1.5%)
Adjuvant chemotherapy	272 (80.5%)
Follow-up period, months (range)	32.16 ± 24.42 (24.07, 1.00–138.17); mean (median)

M; male, F; female, DM; diabetes mellitus, PD; pancreaticoduodenectomy, PPPD; pylorus-preserving pancreaticoduodenectomy, DP; distal pancreatectomy, TP; total pancreatectomy, CA; carbohydrate antigen, FBG; fasting blood glucose, GLR; glucose-to-lymphocyte ratio, NLR; neutrophil-to-lymphocyte ratio, PLR; platelet lymphocyte ratio, AJCC; American Joint Committee on Cancer.

**Table 2 cancers-16-01844-t002:** Preoperative clinical parameters used to estimate disease-specific survival.

	Univariate Analysis		Multivariate Analysis	
Factor	HR (95% CI)	*p*-Value	HR (95% CI)	*p*-Value
Age > 65 years	1.388 (0.891–2.164)	0.147		
Male sex	0.906 (0.579–1.420)	0.667		
DM				
None	1	0.101		
New-onset DM	0.692 (0.436–1.097)	0.117		
Long-standing DM	1.247 (0.851–1.827)	0.258		
Weight loss	1.477 (0.818–2.669)	0.196		
Symptom	1.821 (1.141–2.908)	0.012	1.582 (0.974–2.568)	0.064
Jaundice	1.497 (0.884–2.534)	0.133		
Neutrophil count (×10^6^/mL)	0.981 (0.859–1.120)	0.775		
Platelet count (×10^6^/mL)	1.000 (0.997–1.002)	0.789		
Lymphocyte count (×10^6^/mL)	0.705 (0.506–0.981)	0.038		
Serum glucose (mg/dL)	1.004 (1.001–1.007)	0.007		
GLR > 92.72	2.888 (1.782–4.681)	<0.001	2.475 (1.502–4.078)	<0.001
NLR > 1.06	1.880 (0.775–4.562)	0.163		
PLR > 133.96	1.419 (0.913–2.207)	0.120		
CA 19-9 > 145.35 U/mL	1.994 (1.251–3.180)	0.004	1.577 (0.966–2.572)	0.068
Image size > 2 cm	1.674 (1.072–2.615)	0.024		

HR; hazard ratio, CI; confidence interval, DM; diabetes mellitus, GLR; glucose-to-lymphocyte ratio, NLR; neutrophil-to-lymphocyte ratio, PLR; platelet lymphocyte ratio, CA; carbohydrate antigen.

**Table 3 cancers-16-01844-t003:** A comparison of the clinicopathological characteristics according to GLR.

	Low GLR (*n* = 203)	High GLR (*n* = 135)	*p*-Value
Age, years	62.6 ± 9.97	64.6 ± 9.53	0.094
Sex, M:F	125:78	76:59	0.366
Diabetes mellitus (DM)			<0.001
None	134 (66.0%)	46 (34.1%)	
New-onset DM	35 (17.2%)	41 (30.4%)	
Long standing DM	34 (16.7%)	48 (35.6%)	
CA 19-9, U/mL	271.2 ± 1448.6	824.9 ± 1918.7	0.003
Type of Surgery			0.011
PD/PPPD	102 (50.2%)	87 (64.4%)	
Distal pancreatectomy	96 (47.3%)	42 (31.1%)	
Total pancreatectomy	5 (2.5%)	6 (4.4%)	
Retrieved LNs	17.8 ± 10.9	18.7 ± 12.4	0.487
Tumor size, cm	2.7 ± 1.1	3.1 ± 1.3	0.011
Lymphovascular invasion	76 (37.4%)	60 (44.4%)	0.214
Perineural invasion	148 (73.3%)	102 (75.6%)	0.704
Tumor differentiation			0.659
Well	24 (11.8%)	18 (13.3%)	
Moderate	156 (76.8%)	98 (72.6%)	
Poor	23 (11.3%)	19 (14.1%)	
Lymph node metastasis			0.038
No	100 (49.3%)	51 (37.8%)	
Yes	103 (50.7%)	84 (62.2%)	
pT stage, AJCC 8th			0.058
T1/2	162 (79.8%)	97 (71.8%)	
T3	11 (5.4%)	18 (13.3%)	
T4	30 (14.8%)	20 (14.8%)	
pN stage, AJCC 8th			0.083
N0	100 (49.2%)	51 (37.8%)	
N1	73 (36.0%)	55 (40.7%)	
N2	30 (14.8%)	29 (21.5%)	
R status			0.839
R0	176 (86.7%)	116 (85.9%)	
R1	27 (13.3%)	19 (14.1%)	
Adjuvant chemotherapy	169 (83.3%)	103 (76.3%)	0.114
Recurrence			0.639
Local	33 (29.5%)	20 (24.7%)	
Systemic	56 (50.0%)	46 (56.8%)	
Both	23 (20.5%)	15 (18.5%)	
Complication	123 (60.6%)	83 (61.5%)	0.910

GLR; glucose-to-lymphocyte ratio, M; male, F; female, LN; lymph node, CA; carbohydrate antigen, PPPD: pylorus pre-serving pancreaticoduodenectomy, AJCC; American Joint Committee on Cancer.

**Table 4 cancers-16-01844-t004:** Risk group according to the number of prognostic factors.

Risk Group	N	1-Year Survival Rate	3-Year Survival Rate	5-Year Survival Rate	*p*-Value	*p*-Value
Group 1 (Low-risk)	56	0.91	0.63	0.50	Ref	
Group 2 (Intermediate-risk)	224	0.84	0.46	0.35	0.006	Ref
Group 3 (High-risk)	58	0.71	0.19	0.12	<0.001	<0.001

## Data Availability

The raw data supporting the conclusions of this article will be made available by the authors on request.

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
