# Peer review of "Glucose-to-Lymphocyte Ratio (GLR) as an Independent Prognostic Factor in Patients with Resected Pancreatic Ductal Adenocarcinoma—Cohort Study"

_cancers, 2024, doi:10.3390/cancers16101844_

Round 1

Reviewer 1 Report

Comments and Suggestions for Authors

This article seems to be an interesting investigation.

1. Overall, the presentation of the manuscript can be improved significantly. Text editing needs special attention.

2. A recently published report on the same topic could not be found in the bibliography. Zhang et.al., Int. J Clin. Oncol., 2021 (PMID 32959232) describes GLR in patients with resected pancreatic cancer. This reference was not quoted in this article. How does the current study differ or compare with this published report?

3. The tables and figure legends for tables 4 & 5 and figures 2 & 3 need to be added along with the tables or figures. These have to be explained more clearly in the text.

4. Discussion has to be improved significantly. Please include the references and compare this study with the previous reports.

5. Please increase the resolution of the survival curves.

6. Comments are mentioned in the attached pdf.

Comments on the Quality of English Language

Please improve the text editing such as space between words and review the spelling and grammar with a text editor.

Reviewer 2 Report

Comments and Suggestions for Authors

The author evaluated the impact of glucose-to-lymphocyte ratio (GLR) on the long-term survival for the patients with resectable pancreatic cancer. They analyzed 338 patients who underwent pancreatic resection and they identified GLR as one of the independent prognostic factors.

The topic is interesting for the readership for Cancers, and the sample size appeared adequate. I have a few comments;

1) Because serum glucose level is quite variable regardless of the patient glucose tolerance, I would like to add HbA1c level as reference.

2) They stated they pick the last glucose level prior to the index operation for GLR. How was their management strategy to control preoperative BS? Were they treated with insulin when the patients presented with BS sky high preoperatively? If so, which value was used for GLR, BS at diagnosis (sky high) or BS well controlled with preoperative intervention? In my opinion, it was not safe to proceed pancreatic resection with serum glucose >500.

3) What was the preoperative preparation for patients with jaundice in this cohort? CA19-9 is usually falsely high in those with jaundice/ Thus, they needed to make it clear if the CA19-9 was re-checked after biliary drainage.

Round 2

Reviewer 1 Report

Comments and Suggestions for Authors

Authors have modified the manuscript and it may be accepted for publication.

Reviewer 2 Report

Comments and Suggestions for Authors

The authors revised the manuscript according to my comments. Their responses were reasonable, and the manuscript has improved much.